# Markers of Metabolic Abnormalities in Vitiligo Patients

**DOI:** 10.3390/ijms251810201

**Published:** 2024-09-23

**Authors:** Federica Papaccio, Monica Ottaviani, Mauro Truglio, Andrea D’Arino, Silvia Caputo, Alessia Pacifico, Paolo Iacovelli, Anna Di Nardo, Mauro Picardo, Barbara Bellei

**Affiliations:** 1Laboratory of Cutaneous Physiopathology and Integrated Center of Metabolomics Research, San Gallicano Dermatological Institute, IRCCS, 00144 Rome, Italy; monica.ottaviani@ifo.it (M.O.);; 2Microbiology and Virology, San Gallicano Dermatological Institute, IRCCS, 00144 Rome, Italy; 3Oncologic and Preventative Dermatology, San Gallicano Dermatological Institute, 00144 Rome, Italy; 4Clinical Dermatology, Phototherapy Unit, San Gallicano Dermatological Institute, IRCCS, 00144 Rome, Italy; 5Istituto Dermopatico dell’Immacolata, IDI-IRCCS, 00167 Rome, Italy

**Keywords:** vitiligo, comorbidities, metabolic syndrome, inflammation

## Abstract

While vitiligo is primarily caused by melanocyte deficiency or dysfunction, recent studies have revealed a notable prevalence of metabolic syndrome (MetS) among patients with vitiligo. This suggests shared pathogenic features between the two conditions. Individuals with vitiligo often exhibit variations in triglyceride levels, cholesterol, and blood pressure, which are also affected in MetS. Given the similarities in their underlying mechanisms, genetic factors, pro-inflammatory signalling pathways, and increased oxidative stress, this study aims to highlight the common traits between vitiligo and metabolic systemic disorders. Serum analyses confirmed increased low-density lipoprotein (LDL) levels in patients with vitiligo, compared to physiological values. In addition, we reported significant decreases in folate and vitamin D (Vit D) levels. Oxidative stress is one of the underlying causes of the development of metabolic syndromes and is related to the advancement of skin diseases. This study found high levels of inflammatory cytokines, such as interleukin-6 (IL-6) and chemokine 10 (CXCL10), which are markers of inflammation and disease progression. The accumulation of insulin growth factor binding proteins 5 (IGFBP5) and advanced glycation end products (AGEs) entailed in atherosclerosis and diabetes onset, respectively, were also disclosed in vitiligo. In addition, the blood-associated activity of the antioxidant enzymes catalase (Cat) and superoxide dismutase (SOD) was impaired. Moreover, the plasma fatty acid (FAs) profile analysis showed an alteration in composition and specific estimated activities of FAs biosynthetic enzymes resembling MetS development, resulting in an imbalance towards pro-inflammatory n6-series FAs. These results revealed a systemic metabolic alteration in vitiligo patients that could be considered a new target for developing a more effective therapeutic approach.

## 1. Introduction

Vitiligo represents an acquired chronic disfiguring disease of the skin caused by a progressive loss of functional melanocytes, resulting in depigmented skin macules, often associated with premature whiting or greying of the hairs, eyelashes, and eyebrows [1,2,3]. For a long time, vitiligo was only considered a cosmetic issue. However, it significantly impacts patients’ quality of life due to its pathophysiology [4]. Furthermore, it has an estimated worldwide prevalence of 0.5 to 2%, irrespective of racial preference, and equally affects adults and children of both genders [1,2,3]. There is growing evidence that vitiligo does not merely concern the skin. It appears that vitiligo has the potential to trigger the development of generalized abnormalities, such as glucose intolerance and lipid abnormalities, confirming the systemic nature of the disease [5,6]. The complex vitiligo pathogenesis involves genetic, immunological, autoimmunological, and inflammatory factors [7], which might explain a wide spectrum of its systemic manifestations. It is known that in vitiligo, autoantibodies production results in the development of autoimmunological comorbidities, such as alopecia areata, autoimmune thyroid disease, Addison’s disease, pernicious anemia, type I diabetes mellitus, and myasthenia gravis [3,8]. Furthermore, the rise in these pro-inflammatory cytokines, such as tumour necrosis factor (TNF), interleukin 1 (IL-1), interleukin 6, and other inflammatory factors engaged in vitiligo, are known to be involved in stimulating insulin resistance and atherosclerosis [9]. A study by Karadag et al. found that in vitiligo patients, the high-density lipoprotein (HDL)-cholesterol concentration was decreased, and the LDL/HDL ratio was increased [10]. The same study also reported that even in nondiabetic vitiligo patients, higher insulin resistance and C-peptide levels were observed compared to the control group [10].

Building on the notion that vitiligo may represent a systemic condition manifested through the loss of pigmentation, our vitiligo unit has observed that patients frequently exhibit features associated with metabolic syndrome. Specifically, our findings confirmed elevated fasting blood glucose levels, triglycerides, fatty acid-binding proteins, and LDL cholesterol among these patients [11]. Metabolic syndrome is an alarming worldwide health problem that affects one-quarter of the adult world population [12]. Described as the clustering of abdominal obesity, hypertension, hyperglycemia, and dyslipidemia in an individual, ultimately leading to diabetes mellitus, cardiovascular diseases, or other chronic diseases [13], MetS can result in serious complications, such as myocardial infarction or stroke. Since the pathogenesis of the metabolic syndrome is a well-established condition in some inflammatory skin diseases, including psoriasis [14,15,16], recently, the association between vitiligo and MetS, or some of its components, has attracted the attention of researchers, which further supports that vitiligo not only affects the skin but also that it has several systemic manifestations [17,18,19]. From a genetic point of view, a genome-wide association study identified susceptible loci in vitiligo patients, finding that some of these have a strong association with diabetes mellitus [20]. Many of the risk loci identified are also associated with vitiligo and other autoimmune diseases, proving common molecular pathways among various disorders [20]. It has been shown that melanocytes are also placed in adipose tissue [21,22], taking part in anti-inflammatory reactions and reducing oxygen species [21,22]. Due to their ability to decrease inflammation and oxidative damage, they could prevent metabolic syndrome [22]. Thus, the diminished number of melanocytes and decreased melanogenesis in adipose tissue might be the common reason behind oxidative stress in both vitiligo and MetS [6]. Regarding the pathogenetic multifactor, our group demonstrated that vitiligo is not confined to melanocytes but involves the entire skin and some non-cutaneous cell types, such as peripheral blood cells [23]. We have characterized several metabolic abnormalities in cells from the pigmented skin of vitiligo patients, including impaired mitochondria dynamic functionality reflected in abnormal glucose metabolism and the augmented generation of reactive oxygen species (ROS) [24,25,26]. Therefore, we recently demonstrated that the antidiabetic drug Pioglitazone, a PPARγ agonist, increases the levels of several anaerobic glycolytic enzymes, recovering mitochondrial membrane potential and mitochondrial DNA copy number defects and improving intracellular ATP production in vitiligo cells [27].

Starting from in vitro experiences and based on the above reports, we aimed to elucidate pathogenic common markers sharing between MetS and vitiligo onset in this study. To follow this, we used the serum obtained from 50 subject afferents at our Phototherapy Unit compared with healthy controls. According to our findings from various biochemical approaches, it has been confirmed that vitiligo patients demonstrate a heightened inflammatory profile in comparison to the control group. Our study also revealed imbalanced levels of antioxidant enzymes and abnormalities in LDL cholesterol, folate, and vitamin D levels in these patients. Interestingly, we analyzed cysteine (Cys) and found that it was reduced in vitiligo patients. Mass spectrometry evaluation indicated a modification in lipid composition in vitiligo patients, which is closely related to the inflammatory process. This study would present updated knowledge on potential metabolic disturbances in vitiligo, aiming to clarify that increased awareness of the metabolic aspects is crucial to improving patient care.

## 2. Results

### 2.1. Evaluation of Common Metabolic Syndrome Components

The laboratory tests performed on serum showed that the study groups had low vitamin D and folate levels, as previously reported in the literature [28,29]. No increases in total cholesterol (Chol-Tot) or high-density lipoprotein levels were observed (Figure 1A). In contrast, low-intensity density lipoprotein levels exceeded normal ranges (Figure 1A). Considered as an indicator of cardiovascular disease, homocysteine was elevated in patients with vitiligo [30]. An increased level of homocysteine inhibits tyrosinase and may mediate melanocyte destruction via increased oxidative damage and IL-6 production, triggering autoimmunity and nuclear factor κB activation [31]. In the present analysis, colorimetric quantification of cysteine levels, a metabolite product of homocysteine metabolism, confirmed an inappropriate amount of the above-mentioned amino acid (Figure 1B). Cys is considered a good marker of oxidative stress and a useful tool in diagnosing and monitoring associated pathologic conditions such as cardiovascular diseases, obesity, and insulin resistance [28].

### 2.2. Elevated Inflammatory Markers in the Serum of Vitiligo Patients

A systemic inflammatory state, supported by cytokines that play a central role in the development of vitiligo, has been associated with metabolic syndrome [5,11]. Several studies have shown that the presence of markers of inflammation, such as circulating interleukins, is involved in the mechanism of insulin resistance and other metabolic pathologies, such as atherosclerosis [11,32]. In addition, transcript analyses have shown the overexpression of IL-1, IL-6, and TNF-α in keratinocytes, negatively affecting melanogenesis [33]. It has also been demonstrated that IL-6 and CXCL10 are reliable markers of vitiligo activity status [34]. Aiming to confirm the inflammatory onset, we assessed protein levels using an ELISA assay in the sera of the involved patients. The measurements revealed significantly higher levels of both IL-6 and CXCL10 in vitiligo patients compared to controls (Figure 2A,B). An opposite result was obtained by measuring interleukin-17 (IL-17) levels. As illustrated in Figure 2C, a significantly lower concentration was detected in the vitiligo group compared to the controls. Nevertheless, some conflicting data on the IL-17 levels in vitiligo subjects are detailed [35,36,37,38]. The evidence suggests that IGFBP5 may play a role in the pathogenesis of atherosclerosis, which is the leading cause of cardiovascular disease [39]. In all vitiligo individuals evaluated, we found that IGFBP5 levels were significantly increased compared with controls (Figure 2D). Interestingly, we observed a significantly inverse correlation between IGFBP5 and the cysteine amount (Figure 2E).

### 2.3. Assessment of Oxidative Stress Indicators in Vitiligo

Chronic oxidative imbalance, linked to aberrant production of reactive oxygen species, is a well-known key feature of metabolic syndrome and vitiligo pathophysiology. In vitiligo skin cells, the abnormal production of ROS can originate from mitochondrial dysfunction [24,25,26]. As an antioxidant defence, superoxide dismutase activity lessens oxidative stress and the downstream activation of inflammatory mediators [40,41,42]. In the literature, there are conflicting reports concerning the levels of oxidants and antioxidants, including SOD levels, in vitiligo patients [43,44,45]. In our vitiligo cohort, we measured a substantial reduction in serum SOD activity compared to the control group (Figure 3A). Moreover, the inappropriate activity of SOD was directly correlated to the altered level of Vit D, as highlighted by the scatter plot rating (Figure 3B). However, we did not observe the same outcome regarding catalase activity. As illustrated in Figure 3C, the activity of the measured Cat seems to have a mild decrease in the sera of the reference group. As illustrated in Figure 3C, vitiligo patients presented a mild but significant increment of the Cat activity. Oxidative injury can be considered a crucial step in vitiligo progression [46,47]. Among oxidative biomarkers linked to metabolic syndrome, the release of advanced glycation end-products became of interest in vitiligo comprehension [48]. Our analysis found that the serum levels of AGEs, considered oxidative biomarkers, are significantly higher in vitiligo patients (Figure 3D). This validated result is exciting given that AGE biochemical interactions are one of the major pathways involved in diabetes mellitus, inducing ROS formation, and moving pro-inflammatory cytokines promptly [49,50], which are all joint events between vitiligo and metabolic systemic diseases.

### 2.4. Vitiligo Circulating Fatty Acid Composition

Fatty acids are a fundamental source of energy and important signalling molecules that influence several physiological processes, including inflammation, oxidative stress, and cell membrane composition and properties. Fatty acid metabolism is finely regulated, and imbalances in this process leading to changes in the concentration and/or profile of circulating fatty acids are often associated with chronic pathologies, including inflammatory and autoimmune disorders and metabolic syndrome [51,52,53,54].

To evaluate the possible dysregulation of lipid metabolism in vitiligo, the plasma FA composition of vitiligo patients was analyzed by gas chromatography–mass spectrometry (GCMS) and compared with that of healthy subjects. The data showed, in vitiligo, an overall percentage distribution between saturated (SFAs), monounsaturated (MUFAs), and polyunsaturated fatty acids (PUFAs) like healthy subjects (Figure 4A). Considering the different FAs analyzed and, in particular, the unsaturated fraction more in-depth, we observed a significant increase in γ-linolenic acid (GLA, C18:3n-6) and arachidonic acid (AA, C20:4n-6), both belonging to the FAs of the n6-series (Figure 4B). Linoleic acid (LA, C18:2n-6) and dihomo-γ-linolenic acid (DGLA, 20:3n-6), also included in the n6-series FAs, did not show level variation between vitiligo and healthy samples, except for a limited non-significant reduction and increase, respectively (Figure 4B). Otherwise, the main n3-series FAs, such as α-linolenic acid (ALA, C18:3n-3), eicosapentaenoic acid (EPA, C20:5n-3), and docosahexaenoic acid (DHA, C22:6n3), presented a decreasing trend reaching the statistical significance only for EPA (Figure 4C).

To better describe and understand the data obtained, we evaluated the activity of the enzymes involved in the FA metabolism through specific indexes considering the proper FA product/precursor ratio. In particular, we looked at the n6-series FA biosynthetic pathway, and we observed a significant increase in the delta6-desaturase index (D6D index; GLA/LA, C18:3n6/C18:2n6), which evaluates the C18:3n6 formation starting from C18:2n6, and a significant reduction in the estimated activity of the elongase enzyme converting C18:3n6 to C20:3n6 (Elongase index, C20:3n6/C18:3n6) (Figure 4D). Thus, the increment in the C18:3n6 level found in vitiligo plasma can be explained by the imbalance between the activities of these two enzymes. On the other hand, the delta5-desaturase index, describing the formation of C20:4n6 from C20:3n6 (D5D index; C20:4n6/C20:3n6), showed no difference in the estimated activity between the pathological and non-pathological group (Figure 4D). Despite the unchanged activity of the elongase enzyme compared to healthy controls, higher levels of C20:4n6 are still observed in vitiligo patients, probably due to the increased levels of C18:3n6. Finally, the aggregate desaturase activity index (n6-ADA; C20:4n6/C18:2n6), which considers the overall transformation from C18:2n6 to C20:4n6, was increased, thus indicating a stronger activation of this pathway in vitiligo patients (Figure 4D). Otherwise, a similar evaluation carried out on the estimated enzyme activities involved in the n3-specific pathway revealed a reduction trend, even if not significant, for the n3-aggregate desaturase activity (n3-ADA; C20:3n3/C18:3n3) considering the overall transformation from C18:3n3 to C20:3n3. The imbalance between the n6- and n3-series FA biosynthetic pathways was also highlighted through the increased n6/n3 index given by the C20:4n6/C20:3n3 ratio (Figure 4D).

The overall results indicated in vitiligo an alteration in the FA metabolism with a disproportion in favour of specific n6-series pro-inflammatory FAs, such as C18:3n6 and C20:4n6, together with the reduction in anti-inflammatory n3-series FAs, particularly of C20:5n3. The resulting picture is therefore compatible with a persistent, low-grade inflammatory state in the presence of the pathology.

## 3. Discussion

Despite increasing advances, vitiligo pathophysiology remains elusive, determining an unsatisfactory treatment regime. The challenges that affected people face are not limited to skin symptoms and may also be associated with various other complications. In addition to social or psychological distress, people with vitiligo may be at increased risk for sunburn, eye problems such as iris inflammation (iritis), and hearing loss [55]. Although many mechanisms have been discovered, different etiopathogenic players are still the object of research and analysis today. Considering the in vitro evidence that our group, along with other groups, showed supporting the metabolic impairment in vitiligo onset, we attempted to elucidate the issue of metabolic syndrome in vitiligo patients. Recently, studies have shown an increased risk of developing metabolic syndrome in patients with vitiligo and have considered clinical features of the disease; different groups highlighted the systemic nature of the condition, shining light on the relationship between MetS and vitiligo [6,7,9]. First, evaluating blood parameters, we confirmed an alteration of markers commonly related to metabolic risk factors, such as abnormal LDL fraction cholesterol and reduced folate and vitamin D in the vitiligo cohort. Previous studies indicated a high amount of homocysteine in vitiligo subjects [30,56]. Here, we also evaluated the serum level of cysteine, a downstream product of homocysteine, which is known as a tyrosinase inhibitor and participates in hypertension development, one of the components of metabolic syndrome [57]. The diminished levels of cysteine mirrored an impaired homocysteine production in our patients, suggesting that the vitiligo cohort may be more prone to developing cardiovascular disease. The relationship between MetS and vitiligo includes oxidative stress through the production of ROS, a component of the pathogenesis of both conditions. ROS disrupt mitochondrial function and, in turn, increase glucose levels, intensifying ROS overproduction and leading to morphological changes in mitochondria [58]. A modification trend was observed. Significantly higher levels of IL-6 and CXCL10 were found compared with controls, which aligns with previous reports [34,59]. In addition, the levels of IGFBP5, which are involved in the pathogenesis of atherosclerosis, significantly increased in patients with vitiligo. While it is true that alterations in cytokines concentrations were seen, the level of IL-17 in our study group was substantially lower than the controls. The early literature suggested the role of IL-17 in vitiligo pathogenesis, but a general consensus is still lacking. Several reports have measured a significant increase in IL-17 levels in the skin and blood samples of vitiligo patients [60,61,62]. However, Osman et al., excluding vitiligo patients presenting other autoimmune diseases, reported a non-significant difference in IL-17 amount between patients and controls [63]. The evaluation of the impaired function of the antioxidant enzymes then confirmed the oxidative imbalance. As reported, the activity of SOD was significantly lower in patients than controls. Otherwise, we observed no clear differences in biological catalase activity between the control and the study group. On the contrary, a reduced Cat level, associated with higher SOD antioxidants, was described by other groups [45,64,65,66,67]. For SOD activity, our results were concordant with some earlier studies published [68,69]. The controversial data regarding the levels of antioxidant enzymes reflect the oxidative complexity of vitiligo. Given that vitiligo is a chronic disease characterized by a stable state and acute phases, the opposite issue by different groups could be related to the stage of the disease. Taking that incomplete penetrance, multiple susceptibility loci and genetic heterogeneity characterized the genetics of vitiligo [20,70,71]. The complex genetic scenario described could account for the variability in the expression of different markers that have been evaluated. As part of metabolic syndrome, the prevalence of diabetes mellitus in vitiligo patients was reported in several studies [72,73,74,75]. Various pathogenic mechanisms are thought to be involved in this association, including oxidative stress, free radicals, and growth factor release. It has been shown that chronic hyperglycemia leads to the accumulation of heterogeneous molecules called AGEs [76]. AGEs promote diabetes complications altering extracellular matrix cellular structure, inducing signalling pathways that promote the expression of cytokines and growth factors and stimulate ROS production. The clinical and experimental reports on type II diabetes have focused on AGEs as new biomarkers or therapeutic targets [77,78]. Thus, the increase in the serum AGEs detected in our vitiligo patients further confirmed a common trait between metabolic comorbidities and vitiligo and supported the hypothesis concerning the systemic nature of the disease. Impaired lipid metabolism is considered a driving factor for MetS risk, and an important hallmark associated with its development is the modification in the circulating FA profile [79]. Due to the different properties and functions exerted by FAs, an imbalance in their composition can be related to a disturbance at the level of glucose–insulin homeostasis, oxidative stress, inflammation, and mitochondrial dysfunction [79], all relevant features shared by both MetS and vitiligo. Our findings demonstrate that in vitiligo patients, although the percentage distribution among SFAs, MUFAs, and PUFAs is like that of healthy subjects, there is an alteration in the PUFA composition. The increased levels of γ-linolenic acid (C18:3n6), arachidonic acid (C20:4n6), and, to some extent, dihomo-γ-linolenic acid (C20:3n6) lead to the imbalance towards n6-series FAs, outlining an inflammatory state not balanced by resolving processes. This root low-grade inflammation is further demonstrated by the higher n6/n3 index (C20:4n6/C20:5n3) resulting from the increase in the level of C20:4n6, the major substrate for the synthesis of inflammatory mediators and the decrease in the C20:5n3 responsible instead for the synthesis of pro-resolving mediators such as the resolvins [80,81]. Moreover, considering that PUFAs and the correct n6/n3 ratio affect the homeostasis between glucose and insulin [82,83], their dysregulation observed in vitiligo can partly explain the tendency towards insulin resistance and altered glucose handling [11,27]. Unbalanced PUFA composition results from the alteration in the activity of specific desaturase and elongase enzymes, which are involved in FA metabolic transformations. Modifications found in the enzymatic activity with the consequent changes in circulating FA profile resemble the ones characterizing MetS [84,85,86], to some extent explaining the incidence among vitiligo patients. The grooving interest developing in recent years concerning the systemic nature of vitiligo supports the idea that peculiar depigmentation patches could not be considered just a cosmetic condition. Our findings confirmed that metabolic syndrome markers were more prevalent in vitiligo subjects than healthy controls, rendering vitiligo a skin disease with systemic manifestations. In the present analysis, we aimed to clarify that increased awareness of the metabolic aspects of vitiligo is crucial to improving patient care. It became clear that understanding the comorbidities could be important to elucidate the complex pathogenesis of vitiligo. Further insights are required and could better explain this intricate association.

## 4. Material and Methods

### 4.1. Patients

To analyze the level of markers of metabolic syndromes in patients with vitiligo, 50 patients referred to the Photobiology Unit of the San Gallicano Institute between December 2022 and February 2024 were enrolled. All patients had a clinical diagnosis of non-segmental vitiligo obtained by Wood’s lamp and classified according to the guidelines established by the Vitiligo Global Issue Consensus Conferences (VGICC) [87]. The mean age of the patients, 19 men and 31 women, was 46.3 years. Exclusion criteria for the patient group were vitiligo patients on medications and patients with signs of infection. All included patients did not receive any drugs or other therapeutic options. The control group consisted of age- and sex-matched healthy controls recruited from healthcare staff members. For both groups, there was a slight female dominance. All subjects involved were Caucasian. The study protocol was approved by the institute’s Ethics Commission (Protocol code 751/16, approved 12 January 2016). All participants signed an informed written consent form in agreement with the Declaration of Helsinki principles.

### 4.2. Blood Sampling

Peripheral blood samples of study participants were collected, and serum and plasma were isolated on the same day of blood withdrawal from patients or controls. The samples were separated into aliquots and kept at −80 °C until analysis. Serum samples were evaluated for HDL-cholesterol, LDL-cholesterol, total-cholesterol, folate, and vitamin D using the Roche/Hitachi (Roche Diagnostic S.p.a., Monza, Italy) cobas c 503 according to the manufacturer’s instructions.

### 4.3. Immunoenzymatic and Colorimetric Assays

Among the parameters evaluated in the present study, IGFBP5 was measured by the AVIVA ELISA kit (OKEH00401) as indicated by the manufacturer (AVIVA System Biology Corporation, San Diego, CA, USA).

IL-6 and CXCL10 were evaluated using an ELISA Cohesion kit (CEK1737 and CEK1124) according to the manufacturer’s recommendation (Cohesion Biosciences UK, London, UK). IL-17 serum levels were detected by Human Interleukin 17A ELISA Kit (CSB-E12819h), following the manufacturer indications (CUSABIO, Houston, TX 77054, USA).

The Cat and SOD activity was determined using the Colorimetric Activity Kit (respectively K033-F1 and K028-H1) produced by Arbor Assays (Ann Harbor, MI, USA). Serum levels of advanced glycation end-products were assessed using an ELISA Kit (EH0622) according to Fine Test recommendations (Wuhan, China). Cysteine Colorimetric Assay Kit (E-BC-K352-M) was used to assess serum cysteine under the product’s guidelines (Elabscience, Houston, TX, USA).

All absorbances were spectrophotometrically quantified by DTX 880 Multimode Detector (Beckman Coulter SRL, Milano, Italy).

### 4.4. Extraction and Derivatization of Plasma Fatty Acids

Lithium heparin plasma samples were collected and stored at −80 °C until use. The plasma of the collected blood samples was separated by centrifugation (1800 rpm, 4 °C, 10 min) and was stored at −80 °C until use. After thawing, 50 µL of deuterated palmitoleic acid (d17PA, 80 µM) as internal standard and 500 µL of MeOH were added to the plasma aliquots (50 µL) to perform simultaneous deproteinization of the sample and extraction of fatty acids [88,89]. The resulting samples were mixed by vortexing for 10–15 min and then centrifuged (12,000 rpm, 4 °C, 15 min). The upper organic phase was collected and dried under a nitrogen flow. For analysis by gas chromatography coupled with mass spectrometry (GCMS), the fatty acids were converted to their respective methyl esters (FAME) by acid methylation (500 µL of 2M HCl in MeOH, 80 °C, 1 h). After evaporation under a nitrogen flow, a mixture of water–hexane (1:1) was used for the FAME extraction. The upper organic phase containing FAME was removed, dried under a nitrogen flow, and suspended in isopropanol for subsequent analysis.

### 4.5. GCMS Analysis of Plasma Fatty Acids

Plasma fatty acids were analyzed as FAME by gas chromatography coupled with mass spectrometry (7890A GC gas chromatograph coupled with 5975 VL MS mass analyzer, Agilent Technologies, Santa Clara, CA, USA). Chromatographic separation was performed on an Agilent J&W HP-88 capillary column (30 m × 250 µm × 0.20 µm; Agilent Technologies, Santa Clara, CA, USA) using helium at 1 mL/min as carrier gas. The temperature gradient used was 5 °C/min from 50 °C to 171 °C, followed by a gradient of 4 °C/min up to the temperature of 240 °C. The injector and transfer line temperatures were set at 250 °C and 280 °C, respectively. The temperature of the MS source was set at 230 °C and the quadrupole at 150 °C. Mass spectra were acquired in SCAN mode after electron impact ionization (EI). The identity of the different FAMEs was verified by comparing the retention times and mass spectra with those of a mixture of standards (Supelco, 37 component FAME standard mixture, Bellefonte, PA, USA) and by matching the mass spectra obtained with those present in specific libraries (NIST98). The semi-quantitative analysis was carried out by comparing the response of each analyte with that of the internal standard. Results were expressed as the relative percentage of total FA nmol (nmol%).

### 4.6. Statistical Analysis

Mean and standard deviation were used to express quantitative values of laboratory data. Comparisons between study groups were performed using Student’s *t*-test *p* values of less than 0.05 were regarded as significant. Correlation analysis was conducted to investigate the relationships between various analytes in the dataset. The Spearman correlation coefficient was calculated for all pairs of features in order to accommodate non-normally distributed data. To assess the statistical significance of the observed correlations, *p*-values were computed and subsequently corrected for multiple comparisons using the Benjamani–Hochberg procedure.

## 5. Conclusions

In conclusion, our study underscores systemic metabolic abnormalities in individuals with vitiligo, complementing previous findings of intrinsic cellular metabolic disruptions. Recognizing the link between vitiligo and alterations characteristic of metabolic syndrome could enhance disease management strategies. A limitation of our study is certainly represented by the small sample size of the patient group due to the exclusion of subjects under treatment. Further studies, with a larger sample size and consecutive blood sampling, are needed to strengthen our findings and consequently provide a better interpretation of the data obtained, elucidating the interplay between the various markers common to both vitiligo and metabolic comorbidities.

## Figures and Tables

**Figure 1 ijms-25-10201-f001:**
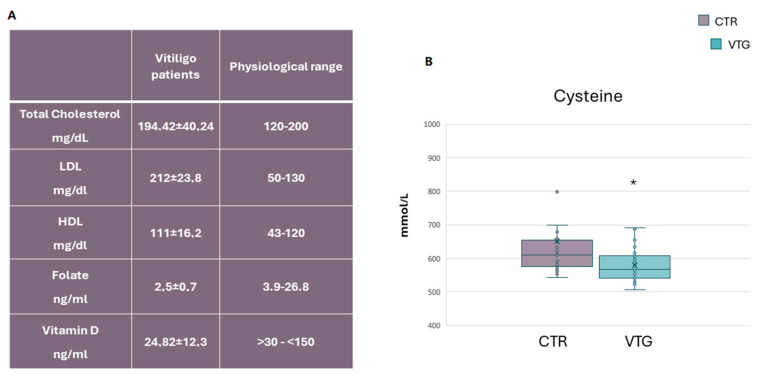
**Laboratory findings of vitiligo patients.** (**A**) Serum metabolic parameters of vitiligo patients compared to the physiological range. Data are represented as mean ± SD. The dot plot summarized the colorimetric determination of cysteine amount in the vitiligo cohort (VTG) compared to the control group (CTR) (**B**). A * *p*-value < 0.05 was considered statistically significant.

**Figure 2 ijms-25-10201-f002:**
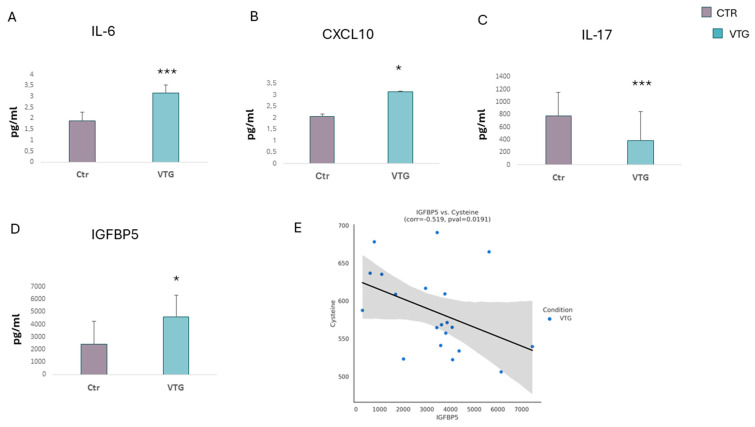
**Inflammatory state evaluation**. Protein detection of several pro-inflammatory markers showed an increase in the level of IL-6 (**A**), CXCL10 (**B**) and IGFBP5 (**D**) in vitiligo patients concerning the controls. In contrast, a decrease in IL-17 (**C**) was measured. (**E**) Correlation between serum level of IGFBP5 and cysteine in patients with vitiligo (n = 50). Data are represented as mean ± SD (* *p* < 0.05, *** *p* < 0.001).

**Figure 3 ijms-25-10201-f003:**
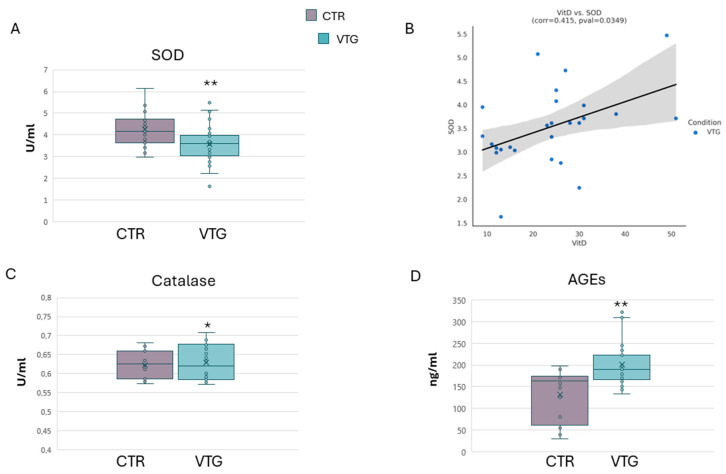
**Serum oxidative biomarkers**. Superoxide dismutase activity was significantly lower among patients than controls (**A**) and correlated with a diminished amount of vitamin D (**B**). As illustrated, a slight catalase rise was detected in the vitiligo cohort compared to the controls (**C**). A higher accumulation of AGEs in vitiligo subjects was reported (**D**). Data are represented as mean ± SD (* *p* < 0.05; ** *p* < 0.01).

**Figure 4 ijms-25-10201-f004:**
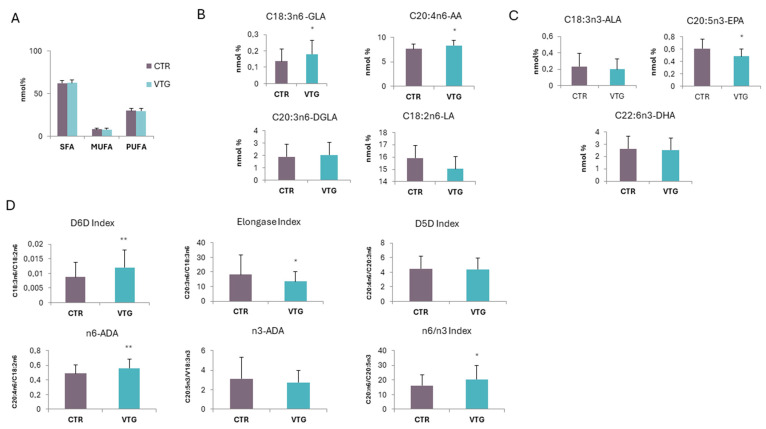
**Circulating Fatty Acids Profile.** GCMS analysis of circulating FAs showed similar SFAs, MUFAs, and PUFAs distribution between vitiligo patients and controls (**A**); vitiligo patients showed alteration in PUFAs composition characterized by an increase in the level of n6-series FAs such as GLA and AA (**B**) and by a decrease in the level of n3-series FAs in particular EPA (**C**). An imbalance in FA metabolic pathway was demonstrated by the alteration observed for the estimated desaturase and elongase activities, with the increase in n6/n3 index (**D**). Data are represented as mean ± SD (* *p* ≤ 0.05; ** *p* ≤ 0.01).

## Data Availability

The raw data presented in this study are available on request from the corresponding author. The data are not publicly available due to privacy (sensible data of both patients and controls with names and surnames).

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
