# Peer review of "Markers of Metabolic Abnormalities in Vitiligo Patients"

_ijms, 2024, doi:10.3390/ijms251810201_

Round 1

Reviewer 1 Report

Comments and Suggestions for Authors

Interesting research study  which revealed a systemic metabolic alteration in vitiligo patients that could  enhance disease management strategies. However, further research is needed to elucidate the interplay between the various markers common to both vitiligo and metabolic comorbidities. Weel designed and thoroughly written. Clinically relevant.

Author Response

Reviewer 1

Interesting research study which revealed a systemic metabolic alteration in vitiligo patients that could enhance disease management strategies. However, further research is needed to elucidate the interplay between the various markers common to both vitiligo and metabolic comorbidities. Weel designed and thoroughly written. Clinically relevant.

We would like to thank the reviewer for his/her kind appreciation of our research.

Reviewer 2 Report

Comments and Suggestions for Authors

Comments to the manuscript ijms-3162821 deals with an interesting study in which the authors evaluate different metabolic markers in samples of patients with vitiligo. A few remarks from the breakdown below

1. it is recommended to check for errors of punctuation, as well as to respect the abbreviations included in the text, and to add the meaning of others also included in the text.

2. Table 1 in subscript A includes at the foot of the figure the meaning of CTG and VTG, and the asterisk

3. the results include means and standard deviations of the results mentioned above (points 2.2, 2.3, and 2.4

4. Justify the sample size

5. The linear results of paragraphs 106 to 110 are suggested to be passed to the materials and methods section

6. Include a weakness paragraph of the study

Author Response

  1. It is recommended to check for errors of punctuation, as well as to respect the abbreviations included in the text, and to add the meaning of others also included in the text.

Accordingly, the entire text has been revised to correct abbreviations and errors of punctuation.

  1. Table 1 in subscript A includes at the foot of the figure the meaning of CTG and VTG, and the asterisk.

As requested, we added the meaning of CTR and VTG in the figure legend and inserted the asterisk.

  1. The results include means and standard deviations of the results mentioned above (points 2.2, 2.3, and 2.4).

In agreement with the reviewer, the information regarding mean±SD was added where necessary.

  1. Justify the sample size.

Thank you for the observation. Here we conducted a retrospective study. Thus, unlike prospective studies, there is no need for sample estimation. Once a study period was defined, we enrolled all patients who attended our Photobiology Unit with a diagnosis of non-segmental vitiligo, who met all the stated inclusion criteria, and who gave informed consent.

  1. The linear results of paragraphs 106 to 110 are suggested to be passed to the materials and methods section.

As requested, we moved the mentioned result, lines 106-110, to the materials and methods part.

  1. Include a weakness paragraph of the study.

We thank the reviewer for this suggestion. We have now discussed this issue in the conclusion part, in lines 333-337.
